# TRIAC Treatment Improves Impaired Brain Network Function and White Matter Loss in Thyroid Hormone Transporter Mct8/Oatp1c1 Deficient Mice

**DOI:** 10.3390/ijms232415547

**Published:** 2022-12-08

**Authors:** Jonathan Rochus Reinwald, Wolfgang Weber-Fahr, Alejandro Cosa-Linan, Robert Becker, Markus Sack, Claudia Falfan-Melgoza, Natalia Gass, Urs Braun, Christian Clemm von Hohenberg, Jiesi Chen, Steffen Mayerl, Thomas F. Muente, Heike Heuer, Alexander Sartorius

**Affiliations:** 1Research Group Translational Imaging, Department of Neuroimaging, Central Institute of Mental Health, Medical Faculty Mannheim, University of Heidelberg, 68159 Mannheim, Germany; 2Research Group Systems Neuroscience and Mental Health, Department of Psychiatry and Psychotherapy, University Medical Center Mainz, 55131 Mainz, Germany; 3Research Group in Silico Psychopharmacology, Central Institute of Mental Health, Medical Faculty Mannheim, University of Heidelberg, 68159 Mannheim, Germany; 4Center for Innovative Psychiatry and Psychotherapy Research, Central Institute of Mental Health, Medical Faculty Mannheim, University of Heidelberg, 68159 Mannheim, Germany; 5Research Group Systems Neuroscience in Psychiatry, Central Institute of Mental Health, Medical Faculty Mannheim, University of Heidelberg, 68159 Mannheim, Germany; 6Leibniz Research Institute for Environmental Medicine, 40225 Düsseldorf, Germany; 7Department of Endocrinology, Diabetes and Metabolism, University Hospital Essen, University of Duisburg-Essen, 45147 Essen, Germany; 8Department of Neurology, University of Lübeck, 23538 Lübeck, Germany; 9Department of Psychiatry and Psychotherapy, Central Institute of Mental Health, Medical Faculty Mannheim, University of Heidelberg, 68159 Mannheim, Germany

**Keywords:** Allan-Herndon-Dudley-syndrome, thyroid hormone, T3, T4, Slc16a2, Slco1c1, brain network, graph analysis, voxel-based morphometry

## Abstract

Dysfunctions of the thyroid hormone (TH) transporting monocarboxylate transporter MCT8 lead to a complex X-linked syndrome with abnormal serum TH concentrations and prominent neuropsychiatric symptoms (Allan-Herndon-Dudley syndrome, AHDS). The key features of AHDS are replicated in double knockout mice lacking MCT8 and organic anion transporting protein OATP1C1 (*Mct8/Oatp1c1* DKO). In this study, we characterize impairments of brain structure and function in *Mct8/Oatp1c1* DKO mice using multimodal magnetic resonance imaging (MRI) and assess the potential of the TH analogue 3,3′,5-triiodothyroacetic acid (TRIAC) to rescue this phenotype. Structural and functional MRI were performed in 11-weeks-old male *Mct8/Oatp1c1* DKO mice (*N* = 10), wild type controls (*N* = 7) and *Mct8/Oatp1c1* DKO mice (*N* = 13) that were injected with TRIAC (400 ng/g bw s.c.) daily during the first three postnatal weeks. Grey and white matter volume were broadly reduced in *Mct8/Oatp1c1* DKO mice. TRIAC treatment could significantly improve white matter thinning but did not affect grey matter loss. Network-based statistic showed a wide-spread increase of functional connectivity, while graph analysis revealed an impairment of small-worldness and whole-brain segregation in *Mct8/Oatp1c1* DKO mice. Both functional deficits could be substantially ameliorated by TRIAC treatment. Our study demonstrates prominent structural and functional brain alterations in *Mct8/Oatp1c1* DKO mice that may underlie the psychomotor deficiencies in AHDS. Additionally, we provide preclinical evidence that early-life TRIAC treatment improves white matter loss and brain network dysfunctions associated with TH transporter deficiency.

## 1. Introduction

Allan-Herndon-Dudley syndrome (AHDS) is a rare X-linked disease with severe neuropsychiatric abnormalities including psychomotor retardation, lack of speech development, dystonia, and severe intellectual deficits. AHDS is caused by inactivating mutations in the monocarboxylate transporter 8 (MCT8), a highly specific thyroid hormone (TH) transporter protein that facilitates the passage of the prohormone 3,3′,5,5′-tetraiodothyronine (T4) and the TH receptor active form 3,3′,5-triiodothyronine (T3) across the plasma membrane [1]. The neuropsychiatric symptoms of MCT8 patients are most likely caused by impaired transport of TH across the blood-brain-barrier (BBB) and, consequently, a profound TH-deficient state in the central nervous system (CNS) that in turn compromises processes such as proper neuronal differentiation, synaptogenesis, and myelination. In addition, patients exhibit highly elevated serum T3 levels that are linked to peripheral symptoms of hyperthyroidism including low body weight, tachycardia, and muscle wasting [2].

The peripheral phenotype of the patients could be fully replicated in *Mct8* knockout (KO) mice, whereas a profound TH deficiency in the murine CNS could only be achieved by a concomitant deletion of *Mct8* and organic anion transporting polypeptide *Oatp1c1*. In rodents, but not in humans or primates, OATP1C1 is highly expressed in brain endothelial cells where it contributes to the T4 transfer across the BBB [3,4,5]. *Mct8/Oatp1c1* double KO (DKO) mice exhibit reduced myelination, decreased thickness of the cortical layers I–IV and pronounced deficits in γ-aminobutyric acid (GABA)ergic interneurons in cortical somatosensory, motor, and cingulate regions, which are hallmarks of impaired TH signaling during brain development [6].

Recently, the TH analogue 3,3′,5-triiodothyroacetic acid (TRIAC) that exerts thyromimetic actions, but is not dependent on MCT8 for cellular entry, has been suggested as a therapeutic option for AHDS [7,8,9]. Indeed, TRIAC induced similar effects as T3 in athyroid mice and positively affected cortical myelination, interneuron maturation, and cerebellar dendritogenesis when administered at high doses between P1 and P12 [10]. A first multicenter trial in humans [8] described the positive effects of TRIAC on peripheral serum T3 levels, body weight, heart rate, blood pressure, and gross motor function in 46 patients with MCT8 deficiency but did not focus on specific effects of TRIAC on neurodevelopment. Of note, a retrospective evaluation of 67 patients who underwent off-label treatment with TRIAC for up to six years confirmed the persistent amelioration of biochemical and clinical parameters but could not address neurodevelopmental outcome due to lack of data [11]. A phase 2 trial (NCT02396459) that specifically aims to address TRIAC action on brain parameters in very young children was initiated only recently and, thus, no outcome has been reported yet. Consequently, investigations of altered brain function in *Mct8/Oatp1c1* DKO mice along with an assessment of TRIAC’s potential to counteract such disturbances in the animal model is even more important.

Non-invasive neuroimaging techniques are a valuable translational tool for investigating alterations in brain structure and function in both patients and animal models [12]. High resolution structural magnetic resonance imaging (MRI) methods such as voxel-based morphometry (VBM) enable the characterization of neuroanatomical differences on a systems level in vivo [13]. Resting-state functional MRI (rs-fMRI), measuring brain activity by detecting changes in blood flow associated with neural activity in a task-negative state, allows for the assessment of functional connectivity (FC) between different brain regions [14]. The implementation of graph analytical methods promoted the view of the brain as a network with multiple levels of organization, characterized by a balance of regional specialization and network integration [15,16]. Specifically, the application of this method to genetically modified animals helped to characterize neuropsychiatric brain phenotypes [17,18,19,20,21]. Potentially, this can establish a mechanistic link between a genetic mutation (i.e., *Mct8/Oatp1c1* DKO), its associated variation (i.e., TH-deficient CNS) and the behavioral phenotype (i.e., intellectual disability). Furthermore, treatment options can be investigated, e.g., by exploring their potential to restore healthy brain network function.

In this study, we provide a first exploration of *Mct8/Oatp1c1* DKO mice with multimodal MRI. With this approach, we aimed to unravel structural and functional brain alterations that may shed further light on the pathogenic mechanisms underlying AHDS. We also investigated the potential of an early-life treatment with the TH analogue TRIAC to normalize altered brain structure and function in *Mct8/Oatp1c1* DKO mice.

## 2. Results

### 2.1. Grey Matter Volume Is Prominently Reduced in Mct8/Oatp1c1 DKO Mice and Cannot Be Rescued by Early-Life Treatment with TRIAC

Immunohistochemical studies demonstrated thinner cortical layers I–IV and alterations in GABAergic interneuron composition in the cerebral cortex of *Mct8/Oatp1c1* DKO mice [6]. We aimed to validate these findings in vivo using non-invasive structural MRI analysis. Specifically, voxel-based morphometry (VBM) revealed significantly reduced grey matter volume in multiple cortical and subcortical brain areas in *Mct8/Oatp1c1* DKO compared to WT mice (*p* < 0.05, FDR-corrected, Figure 1A). The wide-spread cluster covered mostly prefrontal regions including orbitofrontal (OF), infralimbic (IL), prelimbic (PL), cingulate (Cing) and insular (I) cortices, sensorimotor areas (primary and secondary somatosensory cortices, S1, S2; primary and secondary motor cortices, M1, M2) and parts of the striatum (caudate putamen, CPu and nucleus accumbens, NAcc) according to the Allen Brain Atlas (Figure 1B).

In order to evaluate the possible effects of TRIAC treatment on grey matter impairment, we compared *Mct8/Oatp1c1* DKO mice treated with TRIAC during the first three postnatal weeks (*Mct8/Oatp1c1* DKO TRIAC) to the WT group and the untreated *Mct8/Oatp1c1* DKO mice. VBM analysis demonstrated an almost identical pattern of significantly reduced grey matter volume (*p* < 0.05, FDR-corrected) in the *Mct8/Oatp1c1* DKO TRIAC group compared to WT (Figure 1C). Along with the absence of differences between treated and untreated *Mct8/Oatp1c1* DKO mice (*p* > 0.05, FDR-corrected, Figure 1D), these results suggest a lack of treatment effects from TRIAC on grey matter volume.

### 2.2. Mct8/Oatp1c1 DKO Mice Show White Matter Thinning in the Corpus Callosum, Which Can Be Significantly Improved by Early-Life Treatment with TRIAC

Myelination has recently been shown to be severely impaired in *Mct8/Oatp1c1* DKO mice using immunohistological staining [6]. We aimed to corroborate these findings with in-vivo structural MRI applying VBM on white matter tracts of the cerebrum. The comparison of representative structural images of the three groups already indicated a distinct white matter volume reduction in the corpus callosum of *Mct8/Oatp1c1* DKO mice (Figure 2A, middle panel), and further pointed towards a partial compensation of white matter loss by early-life TRIAC treatment (Figure 2A, right panel). For illustration of all individual structural images of the three groups, see Appendix A. Indeed, our quantification of white matter changes with VBM revealed a significant decrease of the volume in *Mct8/Oatp1c1* DKO mice compared to the WT group throughout the whole corpus callosum (Figure 2B, *p* < 0.05, FDR-corrected). Importantly, treatment with TRIAC during the first three postnatal weeks enhanced white matter volume in *Mct8/Oatp1c1* DKO mice. In comparison to untreated *Mct8/Oatp1c1* DKO mice, the TRIAC-treated group showed significantly larger white matter volume especially in the genu and the body of the corpus callosum (Figure 2C, *p* < 0.05, FDR-corrected), while more posterior regions were less affected. However, TRIAC treatment could not fully normalize the white matter reduction in *Mct8/Oatp1c1* DKO mice, as we could still detect differences between WT and *Mct8/Oatp1c1* DKO mice treated with TRIAC (Appendix A, *p* < 0.05, FDR-corrected).

### 2.3. Functional Connectivity Is Strongly Increased within a Broad Cortico-Subcortical Network in Mct8/Oatp1c1 DKO Mice and Can Partially Be Normalized by TRIAC Treatment

Morphological abnormalities including aberrant myelination, grey matter atrophy and altered cortical interneuron marker expression as seen in MCT8 patients and *Mct8/Oatp1c1* DKO mice [6,22,23] may form the structural basis for the neuropsychiatric abnormalities in patients. How these alterations translate into impaired whole-brain network function and subsequently affect behavioral outcome remains largely unknown. In order to fill this important gap, we applied non-invasive rs-fMRI to assess connectivity within functional brain networks. Our network-based statistics (NBS) analysis including 94 brain regions of the Allen Brain Atlas (atlas.brain-map.org, regions illustrated in Figure 1B) provides evidence for a highly significant (*p* < 0.002) increase of functional connectivity (FC) on the whole-brain network level in the *Mct8/Oatp1c1* DKO group compared to WT mice (Figure 3A, primary threshold of p_pt_ < 0.01, NBS permutation test with 5.000 runs and p_NBS_ < 0.025). The significant subnetwork covered 93 brain regions, focusing particularly on prefrontal (PFC), sensorimotor (SM), and temporo-parietal cortices (Temp-Par) as well as striatal and hippocampal regions (HC), and compromised about one ninth of all connections (494 of 4371 connections).

A profound reduction in hyperconnectivity was noted in *Mct8/Oatp1c1* DKO mice treated with TRIAC. The comparison between *Mct8/Oatp1c1* DKO TRIAC mice and *Mct8/Oatp1c1* DKO mice (Figure 3B, primary threshold of p_pt_ < 0.01, NBS permutation test with 5000 runs and p_NBS_ < 0.025) revealed a similar cluster of lower FC (*p* < 0.001) in the treated group, which was only slightly smaller than the previous pattern (410 of 4371 connections), and covered almost the same regions. Of note, no FC cluster survived the NBS analysis between *Mct8/Oatp1c1* DKO TRIAC and WT mice (Figure 3C, primary threshold of p_pt_ < 0.01, NBS permutation test with 5000 runs and p_NBS_ < 0.025), further supporting the pronounced recovery effect induced by TRIAC. Figure 3D shows the treatment-induced functional recovery illustrating the overlapping connections between the significant subnetworks from the *Mct8/Oatp1c1* DKO group compared to the WT group and to the *Mct8/Oatp1c1* DKO TRIAC group, respectively. In brief, these connections demonstrated both hyperconnectivity in the *Mct8/Oatp1c1* DKO mice and normalization of this hyperconnectivity after TRIAC treatment. Specifically, intra- and interhemispheric connections between PFC, striatal, HC, Temp-Par, SM, and thalamic regions (Th/Hyp) were affected by the treatment, while fewer effects could be found in the midbrain areas.

The robustness of our findings of hyperconnectivity in the *Mct8/Oatp1c1* DKO mice and the pronounced treatment effects of TRIAC could be corroborated by the application of a more stringent primary threshold defining the edges for permutation testing in NBS (p_pt_ <0.001, Appendix A). As expected, it only affected the extent of the subnetwork with the major pattern being stable and highly significant (NBS permutation test with 5000 runs and p_NBS_ < 0.025).

### 2.4. Global and Local Brain Network Function Is Strongly Impaired in Mct8/Oatp1c1 DKO Mice and Can Be Restored by Early-Life Treatment with TRIAC

Graph theory has largely contributed to understanding the information flow underlying cognition, behavior, and perception [24]. It conceptualizes the human brain as a network in which anatomical regions represent nodes and correlation coefficients between the regions’ temporal activity form the network’s edges. We used several global and local graph metrics to assess how the increase of FC was observed in *Mct8/Oatp1c1* DKO mice translates into disturbances of healthy brain networks. Most importantly, we could reveal a profound impairment of whole-brain network organization in the *Mct8/Oatp1c1* DKO group by demonstrating a significantly lower small-world index in comparison to WT mice (*p* < 0.05, Figure 4A). Small-worldness characterizes the optimal organization present in healthy brains [21] and is defined as a combination of short path length and high clustering [25]. The shortest path between two nodes is the minimum number of edges that must be traversed to go from one node to another, while the clustering is a measure of a node’s local connection density [25]. Small-world architecture facilitates rapid information transmission and efficient communication within the brain [25], and its disturbance reflects a misbalance between local and long-distance information processing. More specifically, we could find a prominent decrease in local information exchange, quantified by the average local clustering coefficient (*p* < 0.05, Figure 4B), which drove the change in small-worldness. In contrast, characteristic path length, a metric for long-distance information exchange, did not reveal differences between the groups (Figure 4C).

Brain networks also show a modular organization, which is thought to facilitate optimal segregation of information and relative autonomy of function [26]. We found significantly lower modularity in the *Mct8/Oatp1c1* DKO group (*p* < 0.05, Figure 4D), measuring segregated information processing within separated brain modules, thereby corroborating the finding of impaired local information exchange. Furthermore, robustness against targeted attacks, i.e., the network’s ability to maintain an acceptable level of service upon failure of its most important nodes, was significantly lower in *Mct8/Oatp1c1* DKO mice (*p* < 0.05, Figure 4E).

In accordance with our FC findings from NBS, we could observe a substantial effect of the TRIAC treatment on global brain network function. By analogy with the group differences between *Mct8/Oatp1c1* DKO and WT mice, the *Mct8/Oatp1c1* DKO TRIAC group revealed a significantly higher small-world structure (Figure 4A), higher global clustering coefficient (Figure 4B), higher modularity (Figure 4D) and higher robustness against targeted attacks (Figure 4E) for almost all density thresholds and for the respective mean values compared to the untreated *Mct8/Oatp1c1* DKO mice (*p* < 0.05). We found no significant group differences between the *Mct8/Oatp1c1* DKO TRIAC and the WT group, which suggests an extensive recovery of the functional impairments by the early-life treatment with TRIAC.

The results of local graph metrics supported our previous observations of altered network topology, which could be largely normalized by the treatment with TRIAC. We found highly comparable patterns for strength (Figure 5A) and local clustering coefficient (Figure 5B) between WT (left panel) and *Mct8/Oatp1c1* DKO TRIAC mice (right panel), while those of *Mct8/Oatp1c1* DKO mice (middle panel) appeared prominently different. *Mct8/Oatp1c1* DKO mice revealed a significantly higher strength (*p* < 0.05 uncorrected, Figure 5A, middle panel) compared to the other groups, especially in default-mode network regions, including cingulate, prelimbic, retrosplenial, and hippocampal areas. In contrast, local clustering coefficient was significantly lower (*p* < 0.05 uncorrected, Figure 5B, middle panel) in orbitofrontal, motor, striatal, hippocampal and visual areas in *Mct8/Oatp1c1* DKO mice compared to the WT group suggesting impaired regional information processing within these regions. Importantly, treatment with TRIAC induced normalization of strength and local clustering in most of these regions.

## 3. Discussion

Our non-invasive translational MRI study revealed severe impairments of brain structure and function on a systems-level in *Mct8/Oatp1c1* DKO mice, an animal model for AHDS. Previous histomorphological studies already described persistent myelination deficits and disturbed marker expression in cortical GABAergic interneurons, thereby replicating pathophysiological findings observed in patients’ derived brain sections [6,22,23]. Moreover, MRI studies have repeatedly reported myelination deficits in the AHDS brain although it is still a matter of debate whether MCT8 patients display a delayed or permanent hypomyelination [27]. Here, we could confirm white matter thinning in TH transporter deficient animals which is particularly present in the corpus callosum. A recent MRI study demonstrated that such white matter abnormalities in patients with permanent congenital hypothyroidism were associated with worse cognitive function [28]. Moreover, our structural MRI analysis revealed a prominent decrease of grey matter volume in *Mct8/Oatp1c1* DKO mice within a large cluster covering subcortical striatal regions and prefrontal (OF, PL, IL, Cing, I) and sensorimotor cortical areas. These observations fit to patients’ MRI data describing volume loss in cortical as well as in subcortical regions [2]. It is therefore tempting to speculate that in addition to cortical regions, basal ganglia development and function is strongly affected in *Mct8/Oatp1c1* DKO mice. A similar situation may also apply to AHDS patients who frequently present dystonia and hypokinesia as a prominent clinical feature possibly due to disturbed basal ganglia dopaminergic circuit function [2,29,30]. However, whether and how striatal network formation is exactly affected by TH transporter deficiency needs to be further assessed.

Most importantly, our rs-fMRI results provide first evidence for a profound impairment of network function in *Mct8/Oatp1c1* DKO mice. Functional connectivity was found to be significantly increased within a large cluster covering predominantly cortical brain regions and the striatum in TH transporter deficient animals. *Mct8/Oatp1c1* DKO mice also displayed a strong reduction in small-worldness and global network segregation. From a brain network perspective, these disturbances indicate a profound impairment of the optimal balance between efficient information segregation and integration at low wiring and energy costs usually found in the healthy brain [21,31]. We could corroborate this reconfiguration of whole-brain network architecture by demonstrating a decrease of modularity, reflecting impaired integrity of functional brain modules, and a reduced resistance against targeted attacks. Comparable disturbances of global network organization present a common transdiagnostic feature of neuropsychiatric animal models [19,20,21] and neurodevelopmental disorders such as autism [32], schizophrenia [33,34,35,36], and intellectual disability [37,38]. We demonstrate such disturbances for the first time in our AHDS mouse model suggesting them as potential mechanistic background for the neuropsychiatric dysfunctions associated with TH transporter deficiency [39,40].

It is well-established that TH controls cortical GABAergic interneuron development and regulates the expression of distinct calcium binding proteins such as parvalbumin [41]. Thus, not surprisingly *Mct8/Oatp1c1* DKO mice exhibit an altered composition of the cortical GABAergic interneuron network with a highly reduced number of parvalbumin expressing neurons in several regions of the cerebral cortex [6,42]. Furthermore, patch-clamp recordings of cortical neurons revealed an increased frequency of spontaneous miniature inhibitory postsynaptic currents in brain slices of Mct8/Oatp1c1 DKO mice confirming disturbed inhibitory neuronal activity in Mct8/Oatp1c1 deficiency [43]. Whether functional hyperconnectivity and impaired network function detected by rs-fMRI can be fully explained by an aberrant cortical GABAergic neurotransmission, remains to be further investigated particularly as an in-depth electrophysiological examination of the excitatory system is still missing. Hyperconnectivity has been observed for instance in *Tsc2^+/-^* mice with increased mTOR signaling that in turn causes an excitatory synaptic excess putatively due to a disturbed dendritic pruning [44]. Therefore, it will be interesting to assess in future studies whether a similar scenario also applies for *Mct8/Oatp1c1* DKO mice or even for AHDS patients.

Although the underlying molecular cues causing the observed rs-fMRI abnormalities are still not known, functional hyperconnectivity, decreased small-worldness, and reduced global network segregation in *Mct8/Oatp1c1* DKO mice represent a novel disease-characteristic complementing the well-established histomorphological features seen in post-mortem brain sections. As such, they can be considered as a read-out parameter reflecting brain network dysfunction and can be of use in preclinical studies that aim to test and eventually compare the efficacy of different AHDS treatment strategies. As a prominent example for such a preclinical study, treatment of Mct8/Oatp1c1 DKO mice with the thyroid hormone analog TRIAC has been conducted in order to evaluate its potential in replacing TH during critical stages of brain development. When TRIAC treatment was initiated in newborn *Mct8/Oatp1c1* DKO mice and continued during the first three postnatal weeks, normal myelination, undisturbed cerebellar development, and normal cortical GABAergic interneuron maturation could be achieved as shown by immunofluorescence analysis [10,43]. Moreover, this TRIAC treatment protocol was sufficient to restore locomotor performance and to achieve normal electrophysiological activity in cortical brain slices of *Mct8/Oatp1c1* DKO mice [43]. Here, we followed the same TRIAC application protocol and could confirm an improved myelination in TRIAC treated *Mct8/Oatp1c1* DKO animals by structural MRI. Most importantly, we observed a strong normalization of the functional hyperconnectivity and all global graph analytical parameters upon TRIAC treatment suggesting that an early TRIAC application can prevent brain network dysfunction in an AHDS mouse model. In light of these beneficial effects, it remains to be investigated whether AHDS patients receiving a high dose of TRIAC as early as possibly after birth respond to the TRIAC treatment in a similarly positive manner as *Mct8/Oatp1c1* DKO mice.

Despite its beneficial effects on connectivity and myelination, the early-life TRIAC treatment of *Mct8/Oatp1c1* DKO mice was not successful in preventing the grey matter loss in cortical and subcortical areas. The underlying reason for this is unclear and may be related to cell-specific alterations in cellular TRIAC uptake. As a consequence, subsets of neurons may remain “TRIAC”-deficient and show therefore an impaired maturation including reduced dendritic outgrowth. Alternatively, the loss of grey matter volume might be explained by impaired TH transport processes during prenatal stages that in turn will compromise neurogenesis and neuronal migration as seen in other brain areas [42,45]. Furthermore, it is important to note that alterations in cell count and physical tissue volume do not necessarily correlate to grey matter volume changes assessed with VBM. In a recent longitudinal study, a multivariable mixed-effect model incorporating physical tissue volume, cell number, nearest neighbor distance, and nucleus volume estimated with two-photon imaging could only explain 36% of the VBM-estimated grey matter volume variance [46]. The generation and analysis of cell-specific TH transporter deficient mouse models, the application of TRIAC during fetal development, a better understanding of the cellular correlates of VBM measures, and, most importantly, the identification of critical TRIAC transport systems in the CNS are ultimately expected to shed further light on these aspects.

Recently, gene therapy strategies have been developed that include AAV-mediated re-expression of MCT8 in CNS endothelial cells. By this approach, endogenous TH is sought to be transported in sufficient amount across the blood-brain barrier to ensure normal brain development and function. Two different AAV-MCT8 constructs have been tested in *Mct8/Oatp1c1* DKO animals and reported to ameliorate brain abnormalities [47,48]. However, due to different read-out parameters and time points of analysis it still remains to be investigated which of the two vector constructs is more efficient and whether such a gene therapy approach is more or less effective than TRIAC application. For such future studies, we propose to include multimodal MRI imaging to address brain network structure and function in response to the respective treatment regimen.

### Limitations

Histological data and brain TH levels are lacking in our study, making, e.g., correlation analyses between structural MRI findings and histology or between graph metrics and TH levels impossible. Nevertheless, based on the well-established CNS phenotype of *Mct8/Oatp1c1* DKO mice it is reasonable to interpret our data in the context of a profound TH deficient state in the CNS. As VBM analysis are primarily designed to investigate alterations in grey matter, treatment-induced changes in white matter might potentially be underestimated. A further exploration of white matter tracts in the animal model by applying diffusion tensor imaging techniques could expand the investigation to structural connectivity and is certainly of importance [6]. Furthermore, it is elusive whether *Mct8/Oatp1c1* DKO react differently to anesthesia, thereby potentially influencing our functional analysis. However, our profound investigation of physiological parameters argues against group-specific anesthetic effects.

## 4. Materials and Methods

### 4.1. Animals

The generation and genotyping of mice lacking concomitantly MCT8 (Slc16a2^tm1Dgen^; MGI: 3710233) and the organic anion transporting protein OATP1C1 (Slco1c1^tm1.1Arte^; MGI: 5308446) were reported elsewhere [6,49,50]. Heterozygous breeding pairs comprising of Mct8+/- Oatp1c1 fl/- female mice and Mct8 +/y- Oatp1c1 fl/- male mice (all on C57BL/6 background) were set-up to produce the three different experimental groups of male mice used in this study: (1) Wild type (WT) mice (genotype: *Mct8^+/y^ Oatp1c1^fl/fl^* genotype, *N* = 7) as a control group, (2) *Mct8/Oatp1c1* DKO mice (genotype Mct8^-/y^;Oatp1c1^-/-^; *N* = 10), and (3) *Mct8/Oatp1c1* DKO mice daily injected (s.c.) with TRIAC (3,5,3′-L-triioothyroacetic acid; Sigma-Aldrich) at a dose of 400 ng/g body weight between postnatal day 1 and 21 (*Mct8/Oatp1c1* DKO TRIAC, *N* = 13). All animals were supplied with regular chow and water *ad libitum* and housed in IVC cages at a constant temperature (22 °C) and light cycle (12 h light, 12 h dark). Since the MCT8 encoding gene Slc16a2 is encoded on the X-chromosome, only male mice were used in this study.

### 4.2. Study Approval

All animal procedures were performed according to the regulations of animal experimentation within the European Union (European Communities Council Directive 86/609/EEC) and were approved by the Landesamt für Natur-, Umwelt- und Verbraucherschutz Nordrhein-Westfalen (LANUV; Recklinghausen, Germany; AZ 84-02.04.2015.A331) and the Regierungspräsidium Karlsruhe (Karlsruhe, Germany; AZ 35-9185.81/G-51/16).

### 4.3. MRI Acquisition

At the age of 11 weeks, scanning of the animals was conducted at a 9.4 Tesla MRI scanner (94/20 Bruker BioSpec, Ettlingen, Germany) with a two-element anatomically shaped cryogenic mouse surface coil, giving an improvement factor in signal to noise ratio of 2.5 to 3.5 compared to conventional coil setups [51]. Anesthetic regime and recording of the physiological data were performed as previously described [19,51]. Anesthesia was initialized using a mixture of O_2_ (30%) and air/N_2_ (70%) with ~2.5% isoflurane (Baxter Deutschland GmbH, Unterschleissheim, Germany). The depth of sedation was controlled by monitoring body temperature, pO2, heart and breathing rates. The body temperature of the mice was kept in a stable range between 36.5 °C and 37.5 °C using a water heated bed. Breathing rates were monitored using a respiration pad placed beneath the chest (Small Animal Instruments, Stony Brook, NY, USA). Real-time heart and breathing curves were recorded with 10-ms resolution using a signal breakout module (Small Animal Instruments, Stony Brook, NY, USA) and a four-channel recorder (Velleman NV, Gavere, Belgium) together with the scanner trigger pulses for each measured brain volume for later correction of physiological noise.

During structural image acquisition, a 0.2 mL bolus of medetomidine was administered subcutaneously and isoflurane was decreased. Next, a combination of continuous medetomidine (Domitor, Janssen-Cilag, Neuss, Germany, rate 0.2 mg kg^−1^ h^−1^) and isoflurane at 0.1% was used for sedation during rs-fMRI acquisition. High resolution 3D structural images were acquired using a T2-weighted RARE sequence (Rapid Acquisition with Refocused Echoes, RARE factor 16) with a matrix size of 225 × 192 × 96 at echo time (TE) = 62.5 ms and repetition time (TR) = 1.2 s [51]. The rs-fMRI time series were acquired using an echo-planar imaging (EPI) sequence with the following parameters: TR/TE 1300/18 ms, flip angle 50°, 21 slices, 96 × 64 matrix (voxel-size 0.18 mm × 0.18 mm), slice thickness 0.4 mm, 400 acquisitions [51]. As the EPI sequence and the structural scans did not cover the cerebellum sufficiently, this brain area was excluded from the analyses.

### 4.4. Voxel-Based Morphometry (VBM)

VBM analyses were performed using SPM12 software (version 6225, http://www.fil.ion.ucl.ac.uk/spm/software/spm12) and in-house scripts developed in Matlab R2017a (MathWorks Inc., Natick, MA, USA). High resolution structural images were co-registered to a template [19] in Paxinos standard space and brain-extracted [52]. In-house high resolution tissue probability maps [53] were used for the segmentation of the co-registered and brain-extracted original images. Diffeomorphic anatomical through exponentiated algebra (DARTEL) procedure [54] was applied to generate an average-shaped template across all subjects, animal-specific flow field maps representing the transformation of the individual animal map to the average-shaped template, and warped segments reflecting grey matter (GM) and white matter (WM) density. For VBM, modulation of the warped segments was performed to preserve the local GM and WM volume after normalization. The modulated GM and WM images were smoothed with a 0.4 mm Gaussian kernel and entered into a statistical model using age and total intracranial volume as covariates (SPM12). One-way ANOVA with post-hoc two-sample t-tests were used to assess differences between the three groups. The resulting t-maps were spatially corrected for multiple comparisons using false discovery rate (FDR) of *p* < 0.05 in SPM12. The cerebellum was excluded from morphometric analyses due to insufficient image quality.

### 4.5. Preprocessing of rs-fMRI Data

Preprocessing of rs-fMRI data was carried out as previously described including the following steps [19,20,21]. Discard of five initial volumes to avoid influences of magnetization before the scanner achieves steady-state, realignment and unwarping including correction for magnetic field inhomogeneities (SPM12, http://www.fil.ion.ucl.ac.uk/spm/software/spm12), region-specific filtering of heart and respiration rates as typical sources of physiological noise (Aztec software [55]), slice timing correction (SPM12), spatial normalization and co-registration to a mouse brain template in the Paxinos stereotactic coordinate system [56], regression of movement parameters and of the signal from cerebrospinal fluid (FSL, version 5.0.9 http://www.fmrib.ox.ac.uk/fsl). Addressing the common problem of motion artifacts in fMRI data, we additionally integrated a data-driven identification of motion-affected frames based on a decomposition of DVARS, the spatial root mean square of the data after temporal differencing [57,58]. Instead of using an arbitrary threshold to distinguish good from bad scans, the authors suggested a significance test for the DVARS measure, enabling the removal of detected scans based on *p*-values with subsequent scrubbing [58,59]. An exemplary table of our DVARS decomposition is shown in Appendix A and the percentage of scrubbed frames is illustrated in Appendix A. Next, band pass filtering was performed (0.01 to 0.1 Hz, Analysis of Functional NeuroImages (AFNI) software [60]).

Prior to our FC analyses, we performed a comparison of physiological signals and motion parameters, aiming to rule out systematic group effects from nuisance factors such as anesthesia, heart and respiration rate or motion artifacts, which might be associated with, e.g., alterations in peripheral TH levels of the animals. However, no significant group differences could be found between the three groups for heart and respiration rates, framewise displacement and DVARS (Appendix A, *p* > 0.05), supporting a stable and comparable depth of anesthesia, a comparable cardiovascular signal and a similar amount of motion among all groups.

### 4.6. Network-Based Statistic

Mean BOLD time courses were extracted from 94 regions (Figure 1D) based on the Allen Mouse Brain Atlas [61]. Pearson’s correlation coefficients were computed pairwise between all regional time courses for each animal, resulting in individual correlation matrices, representing the FC network of each animal with brain regions forming the nodes and correlation coefficients forming the edges. The functional connectivity matrices were adjusted for age as a covariate and network-based statistic (NBS) was performed to assess FC differences between the three groups. NBS is a non-parametric cluster-based method aiming to identify any potential connected structure formed by a set of suprathreshold links [62]. For post-hoc comparison, two-sample t-tests were performed between the three groups for every single edge of the network. Next, a specific primary threshold was used to discard sub-threshold edges, and the contiguous surviving edges were defined as a cluster. The intensity of the resulting cluster was noted and compared to the maximum intensities of the clusters resulting from 5000 random permutations. Analyses were performed for three different primary thresholds (t_(15)_ > 2.60, t_(18)_ > 2.55 to t_(21)_ > 2.52, corresponding to p_pt_ < 0.01; t_(15)_ > 3.73, t_(18)_ > 3.62, t_(21)_ > 3.53, corresponding to p_pt_ < 0.001), while the alpha-level for the individual permutation test was set to p_NBS_ < 0.025, since a one-sided test was computed within the NBS framework for both directions [63].

### 4.7. Graph-Theoretical Analyses

#### 4.7.1. Network Construction and Density

Network construction and graph metric calculation was carried out as previously described [19,20,21]. For each animal, weighted networks were constructed using the mean time courses from 94 selected regions from the Allen Mouse Brain Atlas [61] which were converted into a digital atlas in Paxinos stereotactic space [64]. All-pairs Pearson’s correlation matrices were normalized by their maximum weights and used to compute FC graphs. Topological characteristics of FC networks were calculated by the computational tools implemented in the Brain Connectivity Toolbox [65] (version 2016-01-16). We focused on a range from 5% to 50% (1% step) network densities for calculating our metrics, using only positive weights. Additionally, we ensured that the networks maintain connectedness, meaning the ability for every node to reach every other node in the network. The range of density thresholds is similar to those of several studies assessing network alterations in humans [33,66] and animals [19,20,21,67]. Mean values over the range from 5% to 50% were calculated to focus the analysis toward identifying systematic effects that are not strongly dependent on a specific threshold level [68].

#### 4.7.2. Graph Metrics for Assessing Regional and Global Architecture

To explore changes in global topology, we calculated small-worldness, global clustering coefficient, characteristic path length [65], modularity and robustness against targeted and random attacks [33].

Brain networks are characterized by functional segregation and functional integration, with the former being the ability for specialized processing within densely interconnected groups of brain regions, and the latter being the ability to rapidly combine specialized information from distributed brain regions [65]. A simple measure of segregation, the clustering coefficient, is equivalent to the fraction of the node’s neighbors that are also neighbors of each other, i.e., the fraction of triangles around an individual node. The most common measure for integration, the characteristic path length, is calculated as the average shortest path length between all pairs of nodes in the network [65]. A well-designed network can combine functionally specialized modules (segregation) with a sufficient number of intermodular links (integration), and is commonly termed as small-world network, with the respective metric being calculated as the ratio between clustering coefficient and characteristic path length [65].

Modular partitions of the individual networks were detected using the Newman algorithm [69,70]. For all density thresholds and animals, the partitioning procedure was run 100 times and the Q-values (ratio between intra-module and extra-module connections) of all runs were averaged. The resulting Q-value represents the modularity for one animal at the given density threshold.

Robustness against random or targeted attack was calculated as the resistance of the network to fragmentation after removal of nodes in either random order or in decreasing order of their degree, respectively [33,71]. Each time a node was removed, global efficiency was recalculated, and robustness was finally computed as the area under the curve of these global efficiency values versus the number of nodes removed [66].

Strength and local clustering coefficient were assessed to explore regional alterations in network structure. Strength was calculated as the sum of weights of edges connected to a given node [67]. Nodes with a large strength have a high total connectivity [65]. Local clustering coefficient quantifies the exchange of information within the vicinity of one node, i.e., the fraction of triangles around an individual node [67,72]. Formulae of all metrics can be found in [65]. Normalization was carried out using comparable random graphs preserving number of nodes, degree distribution, and connectedness as null models [73,74]. Group comparison was performed using one-way ANOVA with post-hoc two-sample t-test on graph metrics adjusted for age as a covariate.

## 5. Conclusions

Our multimodal translational MRI study demonstrates severe impairments of brain structure and function in *Mct8/Oatp1c1* DKO mice that may underlie psychomotor deficiencies in AHDS. Specifically, we reveal wide-spread reductions of cortical and subcortical grey matter volume as well as pronounced white matter thinning in the animal model. Furthermore, *Mct8/Oatp1c1* DKO mice show prominent disturbances of functional brain networks characterized by hyperconnectivity and reduced small-worldness, which indicate a profound impairment of healthy information transport in the CNS. Importantly, we also provide preclinical evidence that early-life treatment with the TH analogue TRIAC improves white matter loss and brain network dysfunctions associated with TH transporter deficiency.

## Figures and Tables

**Figure 1 ijms-23-15547-f001:**
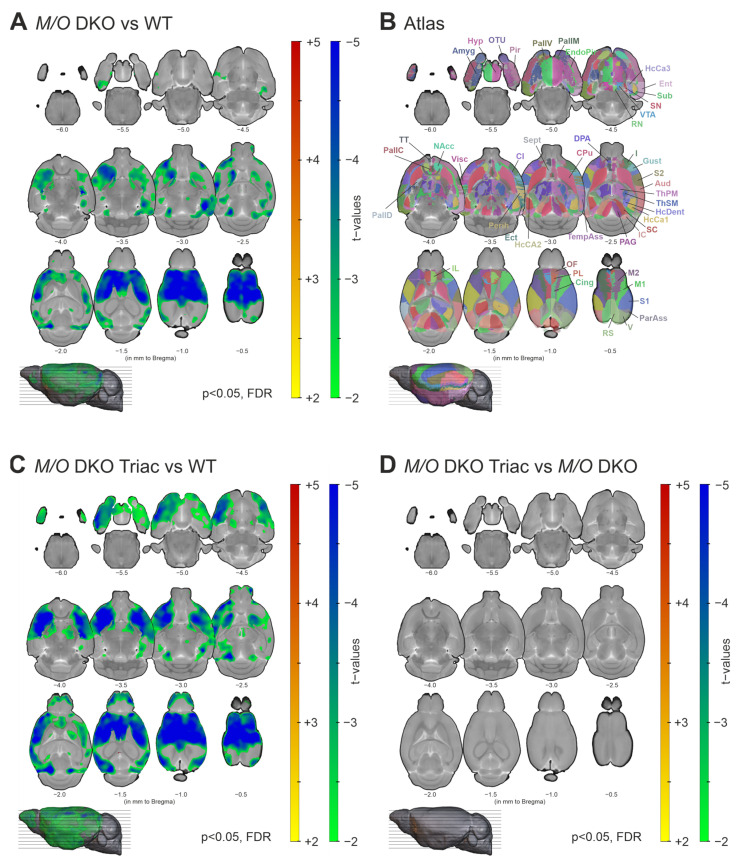
The differences in grey matter volume between WT, *Mct8/Oatp1c1* DKO and *Mct8/Oatp1c1* DKO TRIAC mice. (**A**) Voxel-based morphometry revealed a significant loss of grey matter volume in wide-spread cortical and subcortical areas in *Mct8/Oatp1c1* DKO (*N* = 10) compared to WT (*N* = 7) mice (*p* < 0.05, FDR-corrected). (**B**) Illustrates the atlas regions of the Allen Brain Atlas (atlas.brain-map.org) applied in our study. (**C**) A highly similar pattern of decreased grey matter volume (in relation to **A**) could be found comparing *Mct8/Oatp1c1* DKO mice treated with TRIAC to WT mice (*p* < 0.05, FDR-corrected), while no significant group differences could be found between *Mct8/Oatp1c1* DKO and *Mct8/Oatp1c1* DKO TRIAC (*N* = 13) mice (**D**, *p* > 0.05, FDR-corrected). The results are derived from post-hoc two-sample t-tests and adjusted for age and overall brain size. Grey matter volume reduction in *Mct8/Oatp1c1* DKO mice predominantly focus on prefrontal and sensorimotor cortical areas, the insula and the striatum. Our results indicate a lack of TRIAC treatment effects on grey matter volume reduction. WT, wildtype; *M/O* DKO, *Mct8/Oatp1c1* double knockout; *M/O* DKO TRIAC, *Mct8/Oatp1c1* double knockout mice treated with TRIAC during the first three postnatal weeks; FDR, false discovery error rate correction; IL, infralimbic cortex; PL, prelimbic cortex; Cing, anterior cingulate cortex; M1, primary motor cortex; M2, secondary motor cortex; S1, primary somatosensory cortex; S2, secondary somatosensory cortex; I, insular cortex; Gust, gustatory cortex; Visc, visceral cortex; Aud, auditory cortex; V, visual cortex; RS, retrosplenial cortex; ParAss, parietal association cortex; TempAss, temporal association cortex; Perih, perihinal cortex; Ect, ectorhinal cortex; Ent, entorhinal cortex; EndoPir, endopiriform nucleus; Pir, piriform cortex; TT, taenia tecta; DPA, dorsal peduncular area; HcCA1, hippocampus cornu ammonis 1; HcCA2, hippocampus cornu ammonis 2; HcCA3, hippocampus cornu ammonis 3; HcDent, hippocampus dentate gyrus; Sub, subiculum; Amyg, amygdala; Cl, claustrum; CPu; caudate putamen; NAcc, nucleus accumbens; OTu, olfactory tubercle; Sept, septum; PallC, pallidum caudal; PallD, pallidum dorsal; PallM, pallidum medial; PallV, pallidum ventral; ThPM, thalamus polymodal association cortex related; ThSM, thalamus sensorimotor cortex related; Hyp, hypothalamus; SC, superior colliculus; IC, inferior colliculus; PAG, periaqueductal grey; VTA, ventral tegmental area; SN, substantia nigra; RN, raphe nucleus.

**Figure 2 ijms-23-15547-f002:**
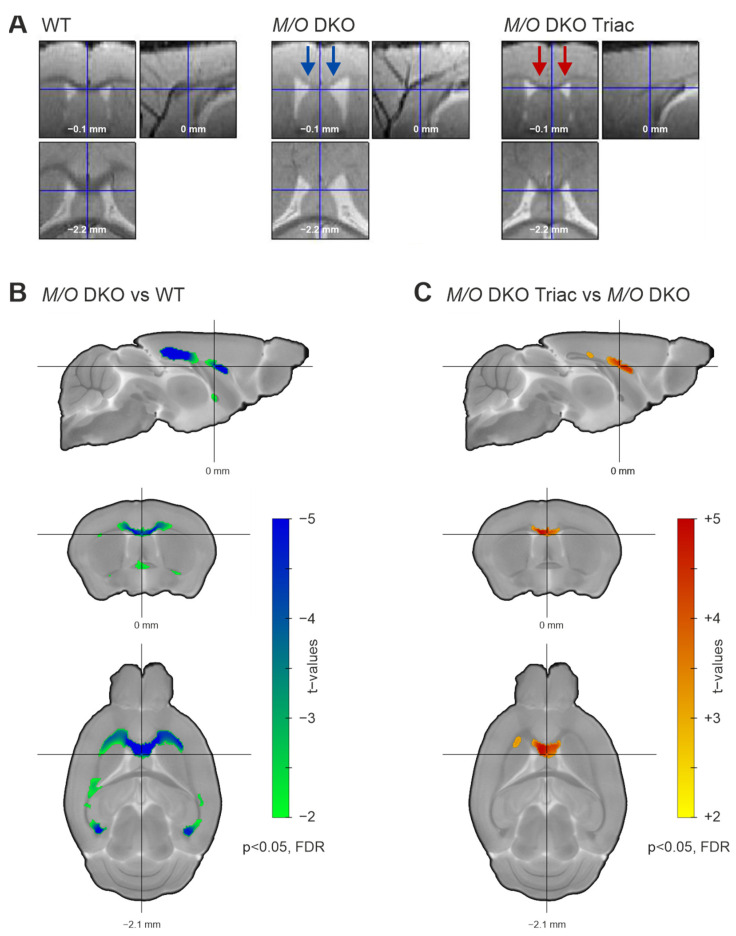
The white matter volume differences between WT, *Mct8/Oatp1c1* DKO and *Mct8/Oatp1c1* DKO TRIAC mice. (**A**) Normalized and warped images of exemplary animals of the three groups (WT, left panel; *Mct8/Oatp1c1* DKO group, middle panel; *Mct8/Oatp1c1* DKO TRIAC, right panel) illustrate white matter volume differences in three orthogonal slices focusing on the corpus callosum. Blue arrows in the middle panel mark white matter volume reduction in *Mct8/Oatp1c1* DKO compared to WT mice visible at first sight. Red arrows in the right panel indicate a partial recovery effect induced by TRIAC treatment in *Mct8/Oatp1c1* DKO. (**B**) Voxel-based morphometry quantifies the loss, demonstrating significantly smaller white matter volume throughout the whole corpus callosum of *Mct8/Oatp1c1* DKO compared to WT mice (green to blue, *p* < 0.05, FDR-corrected). (**C**) The voxel-based morphometry results illustrate a significant recovery effect from TRIAC treatment showing larger white matter volume in *Mct8/Oatp1c1* DKO treated with TRIAC (yellow to red, *p* < 0.05, FDR-corrected) in comparison to the untreated group. White matter volumes were corrected for age and overall brain volume as covariates. WT, wild type; *M/O* DKO, *Mct8/Oatp1c1* double knockout; *M/O* DKO TRIAC, *Mct8/Oatp1c1* double knockout mice treated with TRIAC during the first three postnatal weeks; FDR, false discovery error rate correction.

**Figure 3 ijms-23-15547-f003:**
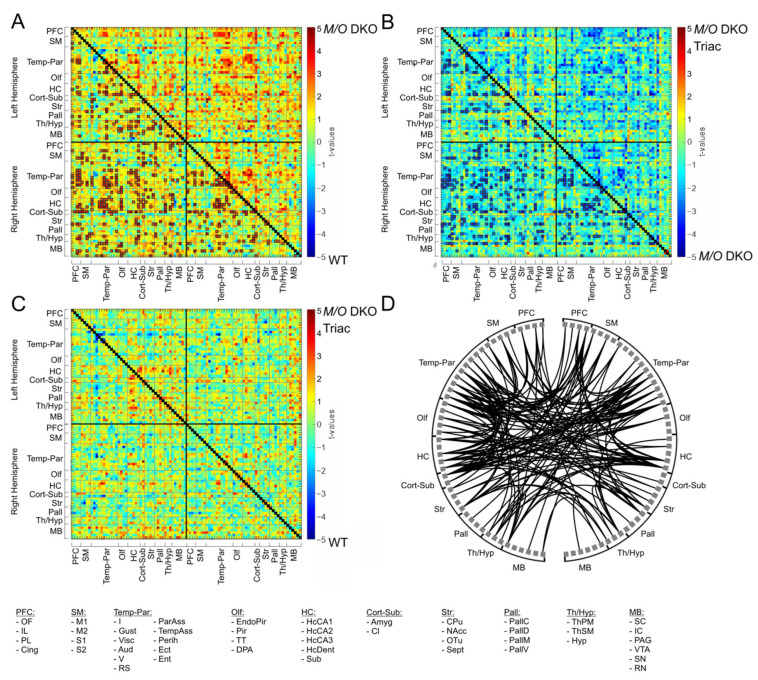
The differences in functional connectivity between WT, *Mct8/Oatp1c1* DKO and *Mct8/Oatp1c1* DKO TRIAC mice. Network-based statistic (NBS) revealed significantly higher functional connectivity between multiple brain regions in *Mct8/Oatp1c1* DKO compared to WT (**A**, orange to red connections marked with black boxes). Early-life treatment of *Mct8/Oatp1c1* DKO mice with TRIAC normalized this hyperconnectivity, demonstrating on the one hand a significantly lower connectivity compared to the untreated *Mct8/Oatp1c1* DKO mice (**B**, light to dark blue connections marked with black boxes) within a pattern covering almost the same regions as in A and on the other hand no significant group differences compared to WT mice (**C**). The schemaball (**D**) indicates the overlapping connections between (**A**,**B**). Thereby, it illustrates the connections that showed both hyperconnectivity in the *Mct8/Oatp1c1* DKO group (compared to WT mice) and normalization of this hyperconnectivity after TRIAC application. In summary, it represents the recovery effect of the treatment with TRIAC in the *Mct8/Oatp1c1* DKO group. NBS results are calculated on functional connectivity matrices adjusted for age as a covariate with a primary threshold of t_(15)_ > 2.52 (**A**), t_(21)_ > 2.60 (**B**), and t_(18)_ > 2.55 (**C**) and a *p*-value for the permutation test (5.000 permutations) of p_NBS_ < 0.025. WT, wild type; *M/O* DKO, *Mct8/Oatp1c1* double knockout; *M/O* DKO TRIAC, *Mct8/Oatp1c1* double knockout mice treated with *TRIAC during the first three postnatal weeks.* For brain regions abbreviations see Figure 1.

**Figure 4 ijms-23-15547-f004:**
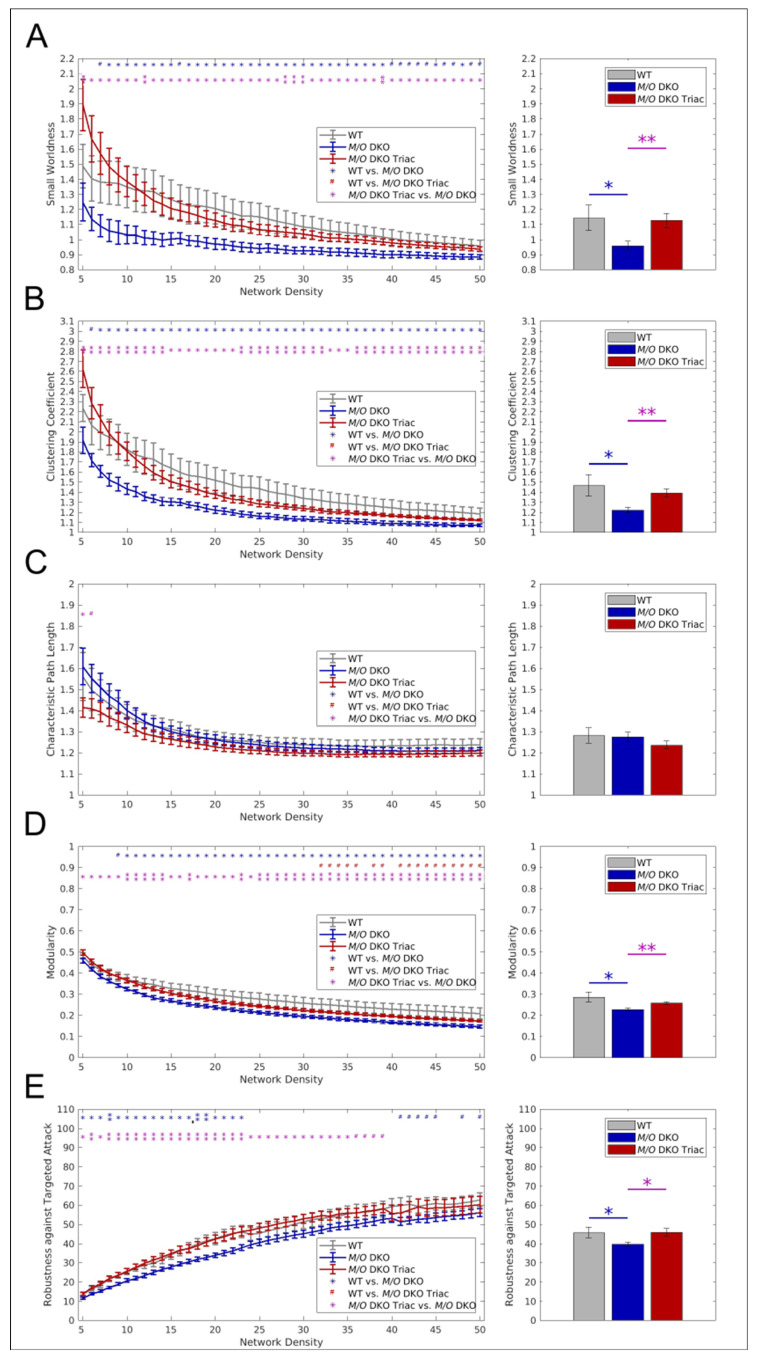
The differences in global graph metrics between WT, *Mct8/Oatp1c1* DKO and *Mct8/Oatp1c1* DKO TRIAC mice. Five global graph metrics are illustrated over network densities from 5% to 50% on the left side. On the right side, mean values calculated over the range from 5% to50% are shown for the same metrics. Significantly lower small-worldness (**A**), global clustering coefficient (**B**), modularity (**D**), and robustness against targeted attack (**E**) could be observed in *Mct8/Oatp1c1* DKO (blue) compared to WT (grey) mice (all *p* < 0.05), while no significant differences could be found for characteristic path length (**C**). Treatment of *Mct8/Oatp1c1* DKO mice with TRIAC during the first three postnatal week (red) almost led to a full normalization of small-worldness (**A**), global clustering coefficient (**B**), modularity (**D**), and robustness against targeted attack (**E**) (all *p* < 0.05). WT, wild type; *M/O* DKO, *Mct8/Oatp1c1* double knockout; *M/O* DKO TRIAC, *Mct8/Oatp1c1* double knockout mice treated with TRIAC. ^#^ *p* < 0.1; * *p* < 0.05, ** *p* < 0.01 (red color: WT vs. *M/O* DKO TRIAC, blue color: WT vs. *M/O* DKO, purple color: *M/O* DKO vs. *M/O* DKO TRIAC).

**Figure 5 ijms-23-15547-f005:**
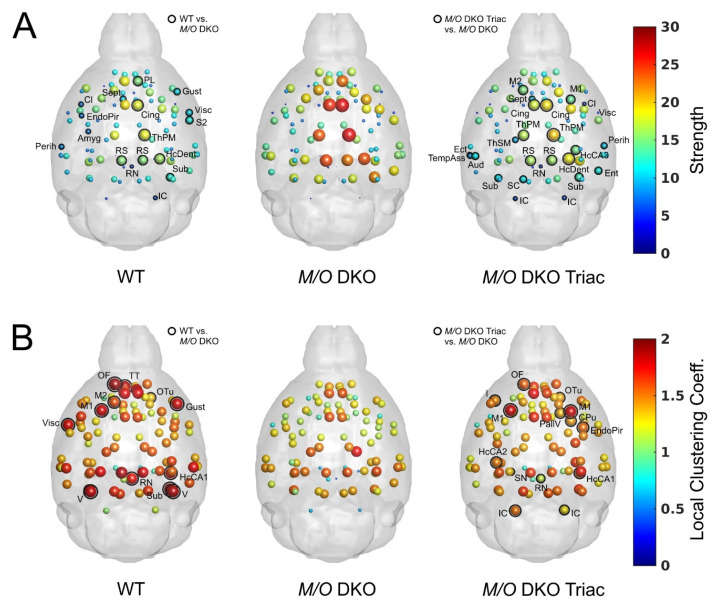
The differences in local graph metrics between WT, *Mct8/Oatp1c1* DKO and *Mct8/Oatp1c1* DKO TRIAC mice. Mean values for strength (**A**) and local clustering coefficient (**B**) calculated over network densities from 5% to 50% are illustrated in the glass brains for WT, *Mct8/Oatp1c1* DKO and *Mct8/Oatp1c1* DKO TRIAC mice, respectively. Of note, the patterns of the two local metrics demonstrated a much stronger similarity between WT (left) and *Mct8/Oatp1c1* DKO mice treated with TRIAC (right) in comparison to the untreated *Mct8/Oatp1c1* DKO group (middle). Significant group differences between WT and *Mct8/Oatp1c1* DKO mice as well as *Mct8/Oatp1c1* DKO TRIAC and *Mct8/Oatp1c1* DKO mice are marked with black rings (*p* < 0.05, uncorrected). The high similarity between the left and the right panel points towards a strong beneficial effect of early-life TRIAC treatment on local graph metrics. WT, wild type; *M/O* DKO, *Mct8/Oatp1c1* double knockout; *M/O* DKO TRIAC, *Mct8/Oatp1c1* double knockout mice treated with TRIAC. For brain regions abbreviations see Figure 1.

## Data Availability

The data presented in this study are available on request from the corresponding author.

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
