# Peer review of "TRIAC Treatment Improves Impaired Brain Network Function and White Matter Loss in Thyroid Hormone Transporter Mct8/Oatp1c1 Deficient Mice"

_ijms, 2022, doi:10.3390/ijms232415547_

Round 1

Reviewer 1 Report

The present study investigates the impact of TRIAC treatment on brain structure and function in Mct8/Oatp1c1 double knockout mice (a model for Allan-Herndon-Dudley Syndrome) using MRI and post-imaging analyses. The study is timely and relevant as treatment strategies for patients are lacking, but clinical trials for TRIAC are ongoing so understanding the potential benefits using brain imaging modalities in rodent models can be useful.

The study reports that TRIAC treatment for the first 21 days after birth ameliorated white matter and connectivity defects observed in Mct8/Oatp1c1 double knockout mice imaged at 11 weeks. However, no improvement the observed reduction in grey matter volume was detected.

Generally, this is very interesting study and perhaps one of the first of its kind, at least with regard to thyroid hormone and brain development. Several minor comments are provided below.

1. Although animal treatments and genotypes are well described, the genotypes of breeding pairs to generate the corresponding genotypes was not described. This should be reported as these transporters may have importance at the placenta.

2. TRIAC treatment clearly had an impact on white matter volume and connectivity in the Mct8/Oatp1c1 double knockout mice. However, reductions in grey matter volume was not improved with TRIAC. This reviewer was curious to understand why a lack of improvement in grey matter might not have occurred considering the robust improvement in other endpoints, but a discussion on the topic was limited. Perhaps expanding the discussion on the topic is worthwhile.

3. The manuscript is of general interest to basic researchers as well as clinicians who have interest in thyroid hormone action in development. However, the description of several of the graph metric endpoints – small worldness, global clustering coefficient, modularity – are difficult to understand for an audience who is not immediately immersed in fMRI analysis field. The overall impact the findings would be better appreciated if these endpoints could be more “simply” defined.

4. More description the Schemaball is necessary to understand what the reader should take from these figures.

5. AHDS is a complex syndrome and as noted in the introduction, it is hypothesized that the associated neurological deficits are the result of CNS hypothyroidism due to the lack of the MCT8 transporter at the blood-brain barrier. Considering this and novelty of using fMRI in this setting, it would be of value to understand to what extent 3 weeks of T3 and/or T4 can restore brain defects in wildtype animals rendered hypothyroid from birth. Although such an experiment may be out of the scope of the present experiment, it may help understand the etiology of the Mct8/Oatp1c1 double knockout mice phenotype and its relationship to CNS hypothyroidism.

Author Response

Response to the Reviewers

Manuscript ID: ijms-2043375

Manuscript Title: TRIAC treatment improves impaired brain network function and white matter loss in thyroid hormone transporter Mct8/Oatp1c1 deficient mice

We express our gratitude to the Editor and the Reviewers for their constructive comments and feedback. We also thank the Editor and the Reviewers for their time and effort in reviewing our manuscript and their thoughtful comments that have been helpful in improving the quality of this paper. Each comment has been carefully considered point by point and addressed in the revised manuscript as follows.

Reviewer #1

Comments and Suggestions for Authors

The present study investigates the impact of TRIAC treatment on brain structure and function in Mct8/Oatp1c1 double knockout mice (a model for Allan-Herndon-Dudley Syndrome) using MRI and post-imaging analyses. The study is timely and relevant as treatment strategies for patients are lacking, but clinical trials for TRIAC are ongoing so understanding the potential benefits using brain imaging modalities in rodent models can be useful.

The study reports that TRIAC treatment for the first 21 days after birth ameliorated white matter and connectivity defects observed in Mct8/Oatp1c1 double knockout mice imaged at 11 weeks. However, no improvement the observed reduction in grey matter volume was detected.

Generally, this is very interesting study and perhaps one of the first of its kind, at least with regard to thyroid hormone and brain development. Several minor comments are provided below.

We sincerely thank the Reviewer for his/her positive comment and the appreciation of the importance of our work.

  1. Although animal treatments and genotypes are well described, the genotypes of breeding pairs to generate the corresponding genotypes was not described. This should be reported as these transporters may have importance at the placenta.

We thank the Reviewer for his/her suggestion and included details regarding our breeding strategy in the “Material and Methods” part.

Material and Methods (line 460-463)

Heterozygous breeding pairs comprising Mct8+/- Oatp1c1 fl/- female mice and Mct8 +/y- Oatp1c1 fl/- male mice (all on C57BL/6 background) were set-up to produce the three different experimental groups of male mice used in this study.

With this breeding strategy, we can rule-out any impact of combined maternal Mct8/Oatp1c1 deficiency on feto/placental growth and rather mimic the situation of MCT8 patients born by heterozygous mothers.

  1. TRIAC treatment clearly had an impact on white matter volume and connectivity in the Mct8/Oatp1c1 double knockout mice. However, reductions in grey matter volume was not improved with TRIAC. This reviewer was curious to understand why a lack of improvement in grey matter might not have occurred considering the robust improvement in other endpoints, but a discussion on the topic was limited. Perhaps expanding the discussion on the topic is worthwhile.

 We thank the Reviewer for raising this important point focusing on the lack of grey matter volume improvement after early-life TRIAC treatment in our study. Following his/her suggestion, we expanded our discussion on this topic incorporating recent histological findings on TRIAC effects and new results from studies comparing 2Pii and VBM, which potentially explain discrepancies between histology and VBM. We added the following paragraph to our “Discussion” section:

Discussion (line 415-432)

Despite its beneficial effects on connectivity and myelination, early-life TRIAC treatment of Mct8/Oatp1c1 DKO mice was not successful in preventing the grey matter loss in cortical and subcortical areas. The underlying reason for this is unclear and may be related to cell-specific alterations in cellular TRIAC uptake. As a consequence, subset of neurons may remain “TRIAC”-deficient and show therefore an impaired maturation including reduced dendritic outgrowth. Alternatively, the loss of grey matter volume might be explained by impaired TH transport processes during prenatal stages that in turn will compromise neurogenesis and neuronal migration as seen in other brain areas (Mayerl et al., 2022, Cells, Mayerl et al., 2021, Cerebral Cortex). Furthermore, it is important to note that alterations in cell count and physical tissue volume do not necessarily correlate to grey matter volume changes assessed with VBM. In a recent longitudinal study, a multivariable mixed-effect model incorporating physical tissue volume, cell number, nearest neighbor distance, and nucleus volume estimated with two-photon imaging could only explain 36% of the VBM-estimated grey matter volume variance (Asan et al., 2021, Scientific Reports). The generation and analysis of cell-specific TH transporter deficient mouse models, the application of TRIAC during fetal development, a better understanding of the cellular correlates of VBM measures, and, most importantly, the identification of critical TRIAC transport systems in the CNS are ultimately expected to shed further light on these aspects

  1. The manuscript is of general interest to basic researchers as well as clinicians who have interest in thyroid hormone action in development. However, the description of several of the graph metric endpoints – small worldness, global clustering coefficient, modularity – are difficult to understand for an audience who is not immediately immersed in fMRI analysis field. The overall impact the findings would be better appreciated if these endpoints could be more “simply” defined.

We thank the reviewer for raising this point concerning the comprehensibility of the description of graph metrics in the manuscript. Indeed, some of the explanations of our metrics were too technically. Therefore, we rephrased the “Results” section reporting the outcomes from graph analysis and provided additional information on the graph metrics.

Results (line 259-286)

Graph theory has largely contributed to understanding the information flow underlying cognition, behavior, and perception [23]. It conceptualizes the human brain as a network in which anatomical regions represent nodes and correlation coefficients between the regions’ temporal activity form the network’s edges. We used several global and local graph metrics to assess how the increase of FC observed in Mct8/Oatp1c1 DKO mice translates into disturbances of healthy brain networks. Most importantly, we could reveal a profound impairment of whole-brain network organization in the Mct8/Oatp1c1 DKO group by demonstrating a significantly lower small-world index in comparison to WT mice (p<0.05, Figure 4A). Small-worldness characterizes the optimal organization present in healthy brains [20] and is defined as a combination of short path length and high clustering (Braun et al., 2016, Neuron). The shortest path between two nodes is the minimum number of edges that must be traversed to go from one node to another, while the clustering is a measure of a node’s local connection density (Braun et al., 2016, Neuron). Small-world architecture facilitates rapid information transmission and efficient communication within the brain (Braun et al., 2016, Neuron), and its disturbance reflects a misbalance between local and long-distance information processing. More specifically, we could find a prominent decrease in local information exchange, quantified by the average local clustering coefficient (p<0.05, Figure 4B), which drove the change in small-worldness. In contrast, characteristic path length, a metric for long-distance information exchange, did not reveal differences between the groups (Figure 4C).

Brain networks also show a modular organization, which is thought to facilitate optimal segregation of information and relative autonomy of function (Bassett et al., 2011, Neuroimage). We found significantly lower modularity in the Mct8/Oatp1c1 DKO group (p<0.05, Figure 4D), measuring segregated information processing within separated brain modules, thereby corroborating the finding of impaired local information exchange. Further, robustness against targeted attacks, i.e., the network’s ability to maintain an acceptable level of service upon failure of its most important nodes, was significantly lower in Mct8/Oatp1c1 DKO mice (p<0.05, Figure 4E).

  1. More description the Schemaball is necessary to understand what the reader should take from these figures.

We thank the Reviewer for this point concerning the shortage of description of the schemaball and its significance for the work. We added the following complementary information to the “Results” section and the figure caption.

Figure 3 (line 241-256)

Network-based statistic (NBS) revealed significantly higher functional connectivity between multiple brain regions in Mct8/Oatp1c1 DKO compared to WT (A, orange to red connections marked with black boxes). Early-life treatment of Mct8/Oatp1c1 DKO mice with TRIAC normalized this hyperconnectivity, demonstrating on the one hand a significantly lower connectivity compared to the untreated Mct8/Oatp1c1 DKO mice (B, light to dark blue connections marked with black boxes) within a pattern covering almost the same regions as in A and on the other hand no significant group differences compared to WT mice (C). The schemaball (D) indicates the overlapping connections between (A) and (B). Thereby, it illustrates the connections that showed both hyperconnectivity in the Mct8/Oatp1c1 DKO group (compared to WT mice) and normalization of this hyperconnectivity after TRIAC application. In summary, it represents the recovery effect of the treatment with TRIAC in the Mct8/Oatp1c1 DKO group. NBS results are calculated on functional connectivity matrices adjusted for age as a covariate with a primary threshold of t(15)>2.52 (A), t(21)>2.60 (B), and t(18)>2.55 (C) and a p-value for the permutation test (5.000 permutations) of pNBS<0.025.

WT, wild type; M/O DKO, Mct8/Oatp1c1 double knockout; M/O DKO TRIAC, Mct8/Oatp1c1 double knockout mice treated with TRIAC during the first three postnatal weeks. For brain regions abbreviations see Figure 1.

Results (line 220-231)

Of note, no FC cluster survived the NBS analysis between Mct8/Oatp1c1 DKO TRIAC and WT mice (Figure 3C, primary threshold of ppt<0.01, NBS permutation test with 5,000 runs and pNBS<0.025), further supporting the pronounced recovery effect induced by TRIAC. Figure 3D shows the treatment-induced functional recovery illustrating the overlapping connections between the significant subnetworks from the Mct8/Oatp1c1 DKO group compared to the WT group and to the Mct8/Oatp1c1 DKO TRIAC group, respectively. In brief, these connections demonstrated both hyperconnectivity in the Mct8/Oatp1c1 DKO mice and normalization of this hyperconnectivity after TRIAC treatment. Specifically, intra- and interhemispheric connections between PFC, striatal, HC, Temp-Par, SM, and thalamic regions (Th/Hyp) were affected by the treatment, while fewer effects could be found in the midbrain areas.

  1. AHDS is a complex syndrome and as noted in the introduction, it is hypothesized that the associated neurological deficits are the result of CNS hypothyroidism due to the lack of the MCT8 transporter at the blood-brain barrier. Considering this and novelty of using fMRI in this setting, it would be of value to understand to what extent 3 weeks of T3 and/or T4 can restore brain defects in wildtype animals rendered hypothyroid from birth. Although such an experiment may be out of the scope of the present experiment, it may help understand the etiology of the Mct8/Oatp1c1 double knockout mice phenotype and its relationship to CNS hypothyroidism.

We fully agree with the Reviewer´s suggestions and would also consider a multimodal MRI analysis of congenital hypothyroid Pax8 KO mice that are postnatally treated with either T4 and/or T3, as a preclinically highly interesting and informative approach. Unfortunately, and as the reviewer already stated, such a time-, work- and cost-intensive animal experiment is out of the scope of our present study as it would take at least three years (including application for novel animal license) to obtain relevant data. Yet, we are eager to address this specific question in the future.

Reviewer 2 Report

The Auhors present extensive MRI investigations on brain structures in the mouse model of AHDS with or without early TRIAC treatment. They show reversibility of several structural abnormalities presumably induced by TRIAC administration thus giving an experimental support to the development of protocols for the treatment in utero of AHDS.

The paper is well written and clearly illustrated and certainly of interest for the general readers of IJMS. I would find appropriate the citation and discussion of:

- van Geest et al, JCEM 2022, who illustrate the beneficial effects of long-term TRIAC treatment in a large real-life study.

- Perri K et al, JCEM 2021, who illustrate functional MRI alterations in children with congenital hypothyroidism in comparison with healthy peers.

Author Response

Response to the Reviewers

Manuscript ID: ijms-2043375

Manuscript Title: TRIAC treatment improves impaired brain network function and white matter loss in thyroid hormone transporter Mct8/Oatp1c1 deficient mice

We express our gratitude to the Editor and the Reviewers for their constructive comments and feedback. We also thank the Editor and the Reviewers for their time and effort in reviewing our manuscript and their thoughtful comments that have been helpful in improving the quality of this paper. Each comment has been carefully considered point by point and addressed in the revised manuscript as follows.

Reviewer 2

Comments and Suggestions for Authors

The Auhors present extensive MRI investigations on brain structures in the mouse model of AHDS with or without early TRIAC treatment. They show reversibility of several structural abnormalities presumably induced by TRIAC administration thus giving an experimental support to the development of protocols for the treatment in utero of AHDS.

The paper is well written and clearly illustrated and certainly of interest for the general readers of IJMS. I would find appropriate the citation and discussion of:

We thank the Reviewer for his/her positive feedback and the appreciation of the relevance of our work.

- van Geest et al, JCEM 2022, who illustrate the beneficial effects of long-term TRIAC treatment in a large real-life study.

- Perri K et al, JCEM 2021, who illustrate functional MRI alterations in children with congenital hypothyroidism in comparison with healthy peers.

We thank the Reviewer for his/her suggestion and implemented both studies in our manuscript.

Introduction (line 79-82)

Of note, a retrospective evaluation of 67 patients who underwent off-label treatment with TRIAC for up to six years confirmed persistent amelioration of biochemical and clinical parameters but could not address neurodevelopmental outcome due to lack of data (van Geest, 2022, J. Clin. Endocrinol. Metab.).

Results (line 346-348)

A recent MRI study could demonstrate that such white matter abnormalities in patients with permanent congenital hypothyroidism were associated with worse cognitive function (Perri et al., 2021, J. Clin. Endocrinol. Metab.).
